# β-Catenin Elicits Drp1-Mediated Mitochondrial Fission Activating the Pro-Apoptotic Caspase-1/IL-1β Signalosome in *Aeromonas hydrophila*-Infected Zebrafish Macrophages

**DOI:** 10.3390/cells12111509

**Published:** 2023-05-30

**Authors:** Shagun Sharma, Manmohan Kumar, Jai Kumar, Shibnath Mazumder

**Affiliations:** 1Immunobiology Laboratory, Department of Zoology, University of Delhi, Delhi 110007, India; 2Faculty of Life Sciences and Biotechnology, South Asian University, Delhi 110021, India

**Keywords:** β-catenin, NOX, mtROS, zebrafish kidney macrophages, *A. hydrophila*, apoptosis

## Abstract

Canonical Wnt signaling plays a major role in regulating microbial pathogenesis. However, to date, its involvement in *A. hydrophila* infection is not well known. Using zebrafish (*Danio rerio*) kidney macrophages (ZKM), we report that *A. hydrophila* infection upregulates *wnt2*, *wnt3a*, *fzd5*, *lrp6,* and β-catenin (*ctnnb1*) expression, coinciding with the decreased expression of *gsk3b* and *axin*. Additionally, increased nuclear β-catenin protein accumulation was observed in infected ZKM, thereby suggesting the activation of canonical Wnt signaling in *A. hydrophila* infection. Our studies with the β-catenin specific inhibitor JW67 demonstrated β-catenin to be pro-apoptotic, which initiates the apoptosis of *A. hydrophila*-infected ZKM. β-catenin induces NADPH oxidase (NOX)-mediated ROS production, which orchestrates sustained mitochondrial ROS (mtROS) generation in the infected ZKM. Elevated mtROS favors the dissipation of the mitochondrial membrane potential (ΔΨ_m_) and downstream Drp1-mediated mitochondrial fission, leading to cytochrome *c* release. We also report that β-catenin-induced mitochondrial fission is an upstream regulator of the caspase-1/IL-1β signalosome, which triggers the caspase-3 mediated apoptosis of the ZKM as well as *A. hydrophila* clearance. This is the first study suggesting a host-centric role of canonical Wnt signaling pathway in *A. hydrophila* pathogenesis wherein β-catenin plays a primal role in activating the mitochondrial fission machinery, which actively promotes ZKM apoptosis and helps in containing the bacteria.

## 1. Introduction

*A. hydrophila* is a Gram-negative, rod-shaped bacterium that is ubiquitous in aquatic environments. The bacterium is predominantly pathogenic to poikilothermic invertebrates and vertebrates, including fish [1]. In fish, it was found to be responsible for the large-scale outbreak of hemorrhagic septicemia, which led to severe economic losses for the aquaculture industry worldwide. *A. hydrophila* is also an important human pathogen causing gastroenteritis and various other systemic infections [2]. The ability to form biofilms, the increasing reports of anti-microbial resistance, and the presence of a wide range of virulence factors coupled with its zoonotic potential makes *A. hydrophila* a pathogen of growing concern [1,3].

*A. hydrophila* pathogenesis involves the concerted activation of numerous signaling cascades [4,5,6,7]. Recent studies have suggested the upregulation of Wnt signaling in *A. hydrophila* infection in several organisms [8,9,10], but the underlying mechanisms remain obscure. Wnt signaling is one of the primordial pathways that evolved in metazoans to execute a myriad of biological functions such as development, regeneration, cell proliferation, motility, and differentiation [11]. It relies on the cognate interaction between Wnt proteins and frizzled (Fzd) receptors, and co-receptors such as low-density lipoprotein receptor-related protein 5/6 (LRP5/6) [11]. Multiple homologs of Wnt proteins and Fzd receptors exist and depending on the type of the ligands involved and downstream events, the Wnt signaling pathways have been broadly divided into the canonical and non-canonical Wnt pathways [12].

Canonical Wnt signaling shuffles between “off” and “on” states. In the “off” state, cytoplasmic levels of β-catenin are maintained at an exceptionally low level by the β-catenin destruction complex composed of adenomatous polyposis coli (APC), axin, casein kinase 1α (CK1α), and glycogen synthase kinase 3 (GSK3), which target β-catenin to proteolytic degradation. The ‘on’ state is initiated once the Wnt ligands bind Fzd and LRP5/6, which stabilizes β-catenin through inhibiting the β-catenin destruction complex [11]. Thereafter, β-catenin translocates into the nucleus and replaces the T-cell factor (TCF) associated co-repressor with the co-activator, resulting in the transcription of the Wnt-associated genes. Thus, the effect of canonical Wnt signaling is dependent on the translocation of β-catenin into the nucleus and inducing the transcription of target genes.

Being a key component of cell and tissue homeostasis, canonical Wnt signaling has also gained recognition as an innate immune attribute, orchestrating interactions between the host immune system and invading pathogens [13]. However, the context-dependent functioning of Wnt signaling makes it difficult to comprehend its role in immune responses. Importantly, there are conflicting reports demonstrating the involvement of canonical Wnt signaling both as a host-driven and pathogen-driven signaling cascade, thereby playing diverse roles at the host–pathogen interface [12,14,15]. While the activation of canonical Wnt signaling was reported in various infections, including *Chlamydia pneumoniae* and *Toxoplasma gondii* [16,17], the opposite was observed in *Pseudomonas aeruginosa* and *Clostridium difficile* [18,19], suggesting its multifaceted role in regulating immune responses.

Amongst the several downstream factors that β-catenin regulates, reactive oxygen species (ROS) generation is an important phenomenon [20], although the exact mechanism underlying this process remains elusive. On the other hand, there are also reports suggesting that oxidative stress can lead to the activation of β-catenin-dependent signaling events [21]. ROS play an essential role in proteostasis and host immune defenses against diverse pathogens [22]. However, excessive ROS has been linked with the principle of oxidative stress and creating a micro-environment that favors apoptosis [23,24].

Several reports have reviewed the role of mitochondria as a central hub in innate immunity [25]. Among the different alarmins regulated by this organelle, mitochondrial ROS (mtROS) are important. When produced at normal levels, mtROS play an important role in the development of the antimicrobial response [26], but elevated mtROS drives inflammatory responses in microbial pathogenesis [27,28]. Interestingly, NOX-derived ROS acts as a positive regulator of mtROS generation [29], though the molecular underpinnings remain unclear. There are also reports suggesting that activated canonical Wnt signaling acts as a modulator of both mtROS and inflammation [30]. To the best of our knowledge, studies linking canonical Wnt signaling with mitochondrial dysfunction and pro-inflammatory responses have not been undertaken in *A. hydrophila* pathogenesis.

Mitochondrial dynamics are the balance between mitochondrial fusion and fission, which is paramount to maintaining cell fate. Mitochondrial dynamics are regulated by intricate signaling that involves the pro-fission molecule dynamin-related protein1 (Drp1) and the pro-fusion molecules, mitofusin 1 and 2 (Mfn1 and Mfn2, respectively) [31]. It has been suggested that mtROS overproduction favors fission by increasing *dnm1l* expression and inhibiting *mfn* expression [31,32]. However, there are also reports implicating canonical Wnt signaling prompts Drp1-mediated mitochondrial fission [33], but the molecular intermediates assisting this are not yet known. Also, little is known about the downstream signaling components associated with *A. hydrophila*-induced mitochondrial fission.

We previously reported that *A. hydrophila* infection leads to functional alterations in the mitochondria with pro-apoptotic implications [34]. To further extend our findings, we subsequently reported the role of mtROS in altering mitochondrial permeability, thereby causing apoptosis of *A. hydrophila*-infected macrophages [35]. Although there are a few studies which assessed the correlation between canonical Wnt signaling and apoptosis [36,37], the same has not been studied in *A. hydrophila* pathogenesis.

Zebrafish kidney contains both immune and renal cells, and ZKM are inherently phagocytic and serve as an attractive model to study bacterial infections [38]. In the present study, we have used ZKM to study the role of canonical Wnt signaling in pathogenesis induced by *A. hydrophila*. Our findings establish an intriguing axis of β-catenin-ROS-mtROS–Drp1 that culminates in ZKM apoptosis promoting *A. hydrophila* clearance.

## 2. Materials and Methods

### 2.1. Ethics Statement

Zebrafish experiments were conducted following the guidelines set forth by the Animal Ethics Committee, University of Delhi (DU/ZOOL/IAEC-R/2013/32) and were performed according to the protocols approved by the Committee for the purpose of Control and Supervision of Experiments on Animals (CPCSEA), Govt. of India.

### 2.2. Animal Care and Maintenance

Selectively bred zebrafish born from one mother were kept in the Animal House Facility, University of Delhi at 28–30 °C under a 14 h light/10 h dark diurnal photoperiod. Prior to setting up the experiments, fish (6–7 months old) were acclimatized to laboratory conditions for 15 days. Throughout the study, fish health was monitored regularly, and they were maintained as described earlier [39].

### 2.3. Bacterial Culture

*A. hydrophila* (strain 500297) used in the study was gifted from Dr. T. Ramamurthy, the National Institute of Cholera and Enteric Diseases, India. The pathogenicity of the isolate was checked, and the bacteria were grown to log phase in brain–heart infusion broth (pH 7.4, HiMedia, Kennett Square, PA, USA) containing 100 µg/mL ampicillin overnight at 30 °C with proper aeration [34].

### 2.4. ZKM Isolation

Zebrafish were sacrificed using an excess of MS222, and the kidney was aseptically removed and placed in sterile PBS. Single-cell suspensions were prepared by homogenizing the kidney using an electric homogenizer for 30 s. The cell suspension was centrifuged at 400× *g* for 10 min at 4 °C, with the supernatant discarded, the pellet re-suspended and layered on a discontinuous percoll density gradient (34/51%), and centrifuged at 4000× *g* for 10 min at 4 °C. The phagocyte-rich fraction was collected, washed, and left for overnight adherence in 24-well culture plates at 30 °C in a CO_2_ incubator. The non-adherent cell fraction was discarded the following day, and the macrophages were collected by washing with chilled RPMI-1640 (Gibco, Waltham, MA, USA). ZKM purity was determined by flow cytometric analysis and staining with Wright Giemsa (90%), and the percentage viability (>90%) was determined by the 0.4% trypan blue dye exclusion method.

### 2.5. Infection Studies

For infection, ZKM were first washed in antibiotic-free RPMI 1640 supplemented with 10% FBS (Gibco-Invitrogen) and were then infected with *A. hydrophila* grown to log phase at different multiplicities of infection (MOIs) (ZKM: bacteria). A short spin of 5 min was given to facilitate bacteria–ZKM interactions, which were then distributed in 24-well culture plates and incubated for 1 h at 30 °C in the presence of 30 µg/mL chloramphenicol [34]. The concentration of chloramphenicol effectively killed extracellular bacteria without affecting ZKM viability. Finally, the infected ZKM were washed and re-suspended in RPMI-1640 supplemented with 10% FBS, 25 mM HEPES, and 1% penicillin streptomycin (complete-RPMI) for further studies.

### 2.6. Reagents and Chemicals

ZKM were pre-treated separately with β-catenin inhibitors (JW67, 5 µM, Sigma, St. Louis, MI, USA; and MSAB, 10 µM, MedChemExpress, Monmouth Junction, NJ, USA), NADPH oxidase (NOX) inhibitors (Apocynin [Apo], 100 µM, Sigma; and Diphenyleneiodonium chloride [DPI], 10 µM, Sigma), Drp1 inhibitor (Mdivi-1, 25 µM, Sigma), Mitochondrial permeability transition pore (MPTP) inhibitor (Cyclosporin A [CsA], 5 µM, Sigma), mtROS inhibitor (YCG063, 10 µM, Calbiochem, Darmstadt, Germany), caspase-1 inhibitor (Z-YVAD-FMK, 7.5 µM, Sigma), and caspase-3 inhibitor (Ac-DEVD-CHO, 10 µM, Biovision, Waltham, MA, USA), respectively, for 1 h prior to addition of the bacteria to the ZKM culture. Table 1 summarizes the list of chemicals used in the study along with their functions (Table 1). The doses were selected based on their specificity and cytotoxicity. The ZKM treated with the indicated concentrations of the inhibitors remained as viable as control ZKM at all-time points as determined by the trypan blue dye exclusion method and were maintained throughout the experimental timeline.

### 2.7. Apoptosis Studies

The ZKMs pre-treated with or without the indicated inhibitors, or transfected with sc-siRNA or *l1b*-siRNA were infected with *A. hydrophila,* and at indicated time points the resulting apoptosis was studied with various methods.

**(i) DNA ladder assay:** ZKM (5 × 10^6^) were lysed at 24 h post infection (p.i.) using lysis buffer containing 0.2% Triton X-100, 10 mM Tris (pH 7.2), 1 mM EDTA (pH 8.0), and proteinase K (5 mg/mL). The genomic DNA was extracted, electrophoretically separated on 1.8% agarose gel along with 100 bp DNA ladder (Roche) and visualized under the UV transilluminator after staining with 0.5% ethidium bromide [7].

**(ii) Hoechst 33342 staining:** ZKM (1 × 10^6^) were collected at 24 h p.i., washed, and fixed in 3.7% paraformaldehyde. The fixed ZKM were then stained with Hoechst 33342 (2 μg/mL, Sigma), mounted with fluoroshield (Sigma), and visualized under a fluorescence microscope (×40, Zeiss Imager, Jena, Germany, Z2). Three different fields having at least 100 cells were observed for determining the percentage of Hoechst-positive ZKM [7].

**(iii) Annexin V-Propidium Iodide (AV-PI) staining:** ZKM (1 × 10^6^) were collected at 24 h p.i., stained with AV-PI following the manufacturer’s protocol (BD Pharmingen), washed, mounted with fluoroshield, and visualized under a fluorescence microscope (×40, Zeiss Imager, Z2). Three different fields having at least 100 cells were observed for determining the percentages of AV^+^PI^+^, AV^−^PI^+^, and AV^+^PI^−^ cells, respectively [7].

### 2.8. RNA Isolation, cDNA Synthesis and RT-qPCR

ZKM (2 × 10^6^) pre-treated with or without the indicated inhibitors were infected with *A. hydrophila* and at indicated time points, ZKM were collected in TRI reagent (Sigma-Aldrich Corp., Darmstadt, Germany). The total RNA was isolated following the manufacturer’s protocol. After checking the purity and integrity of RNA, cDNA was synthesized using a first-strand cDNA synthesis kit (Thermo Fisher Scientific, Waltham, MA, USA) from 1 μg of DNase-treated (RNase-free) RNA. Diluted cDNA (1 µL) was used as a template for RT-qPCR using the SYBR Green PCR Master Mix (Applied Biosystems, Foster City, CA, USA). The conditions for amplification were incubation at 95 °C for 10 min, followed by 40 cycles at 95 °C for 15 s and at 60 °C for 1 min, and one cycle at 95 °C for 15 s, at 60 °C for 1 min, and finally at 95 °C for 15 s (ViiA7, Applied Biosystems, Foster City, CA, USA). The comparative ΔΔC_T_ method was used to determine the expression levels of the analyzed genes, and relative fold changes were normalized against the housekeeping gene, *actb1* [39]. Table 2 enlists the sequences of real-time primers used in this study.

### 2.9. siRNA Transfection

siRNA transfection was performed using the HiPerFect transfection reagent (Qiagen, Valencia, CA, USA) as per the manufacturer’s instructions. Briefly, 5 µL scrambled siRNA (sc-siRNA) or *il1b*-siRNA, and 5 µL HiPerfect were added to 90 µL Opti-MEM (Invitrogen) and incubated for 20 min at 30 °C to facilitate the siRNA transfection complex formation. Next, ZKM (2 × 10^6^) were gently mixed with the transfection complex and incubated at 30 °C in the presence of 5 % CO_2_ for 16 h, washed, and then infected with *A. hydrophila*. The efficiency of knockdown was confirmed by RT-qPCR analysis (Appendix A). The list of siRNA used in the study along with itssequence are mentioned in Table 3.

### 2.10. Intracellular Bacterial Quantification

ZKM (1 × 10^6^) pre-treated with or without the indicated inhibitors, or transfected with sc-siRNA or *il1b*-siRNA were infected with *A. hydrophila.* At indicated time points p.i., the ZKM were lysed (0.1% Triton X-100), following which 20 µL MTT (3-(4,5-dimethylthiazol-2-yl)-2,5-diphenyltetrazolium bromide, 5 mg/mL, Merck) was added, and the lysate was then incubated for 5 h at 30 °C. The formazan crystals were dissolved in 100 µL dimethyl sulfoxide (DMSO, Sigma) and the absorbance was read at A_595_. The number of intracellular bacteria was enumerated by interpolating the obtained absorbance values on the standard curve [40].

### 2.11. β-Catenin Assay

β-catenin concentration was measured using the CTNNβ1 ELISA Kit (Elabscience) using the chemicals that were supplied with the kit. Briefly, ZKM (2 × 10^6^) pre-treated with or without the indicated inhibitors were infected with *A. hydrophila*, and 100 µL of the cell-free culture supernatant collected at indicated time points p.i. were added to each well and incubated for 12 h at 4 °C. The wells were then washed, and 100 µL of biotinylated detection antibodies (1:100) were added to each well and incubated for 60 min at 37 °C. The plates were washed in 1× wash buffer and 100 µL of HRP-conjugated secondary antibodies (1:100) were added to each well and incubated at 37 °C for 30 min. Following washing with 1× wash buffer, 90 µL of substrate was added and further incubated for 15 min at 37 °C. Finally, 50 µL of stop solution was added to each well, and the absorbance was read at A_450_ using a microplate reader (Epoch2, BioTek, Bucheon, Republic of Korea). The β-catenin protein concentration was enumerated by interpolating the values on the standard curve.

To obtain the nuclear fraction, ZKM (2 × 10^6^) were re-suspended in 500 µL of fractionation buffer (20 mM HEPES [pH 7.4], 10 mM KCl, 2 mM MgCl_2_, 1 mM EDTA, 1 mM EGTA, 1mM DTT, and protease inhibitor cocktail), passed through a 27-gauge needle, and centrifuged at 720× *g* for 5 min. The nuclear pellet was then washed with the fractionation buffer, passed through the needle, and centrifuged at 720× *g* for 10 min. The supernatant was discarded and the pellet was re-suspended in Tris-buffered saline containing 0.1% SDS and then sonicated for 10 s in ice. Thereafter, the β-catenin levels were measured as described above.

### 2.12. Superoxide Measurement

ZKM (1 × 10^6^) pre-treated with or without the specific inhibitors were infected with *A. hydrophila*. The ZKM were collected at indicated time points and superoxide ion measured using nitroblue tetrazolium (NBT, 1 mg/mL, Sigma). Briefly, the cell pellet was resuspended in 100 µL NBT and incubated for 5 h at 30 °C. The pellet was then dissolved in 70 µL DMSO (Sigma) and 60 µL KOH (2 M). The absorbance was read at A_620_ using a microplate reader [34].

### 2.13. mtROS Production

ZKM (2 × 10^6^) pre-treated with or without the indicated inhibitors were infected with *A. hydrophila*. The ZKM were collected at indicated time points p.i., washed, and incubated with 5 µM MitoSOX^TM^ Red mitochondrial superoxide indicator (Molecular Probes) for 20 min in the dark. The pellet was resuspended in 100 µL 1× PBS, and the changes in fluorescence intensity were quantified at an excitation/emission of A_510_/A_580_ using a multi-mode microplate reader (BMG Labtech, Ortenberg, Germany) [35].

In a parallel study, ZKM treated with MitoSOX^TM^ were washed, and the pellet was resuspended in 100 µL 1× PBS and incubated with DAPI (100 μg/mL, Sigma) for 15 min. The excess dye was washed and the ZKM were then mounted on a slide with fluoroshield, and three different fields were visualized using a fluorescence microscope (×40, Zeiss Imager, Z2).

### 2.14. Mitochondrial Membrane Potential (ΔΨ_m_) Assay

The changes in ΔΨ_m_ were studied using rhodamine 123 dye (Molecular Probes) [34]. Briefly, ZKM (2 × 10^6^) pre-treated with or without the indicated inhibitors were infected with *A. hydrophila*. ZKM were collected at indicated time points p.i., washed, and incubated with rhodamine 123 (10 µM) for 30 min at 37 °C in dark. Excess dye was removed with repeated washing, and the fluorescence was recorded at excitation/emission of A_511_/A_534_ (BMG Labtech).

### 2.15. Measurement of Cyt c

The cyt *c* release was measured according to the mentioned protocol [41]. Briefly, ZKM (1 × 10^6^) pre-treated with or without the indicated inhibitors were infected with *A. hydrophila*. The cultures were terminated at indicated times p.i., homogenized in buffer A (50 mM Tris, 1 mM PMSF, 2 mM EDTA, pH 7.5), followed by the addition of 2% glucose, and centrifuged at 2000× *g* for 10 min at 4 °C. The supernatant was treated with ascorbic acid (500 mg/mL) for 5 min and absorbance was read at A_550_ to detect the release of cyt *c* in the cytoplasm.

### 2.16. Mitochondrial Morphology Assessment

The mitochondrial morphology was analyzed with confocal microscopy using MitoTracker Green dye (Molecular Probes). ZKM (1 × 10^6^) pre-treated with or without the indicated inhibitors were infected with *A. hydrophila*, and cultures were terminated at 12 h p.i. The ZKM were incubated with MitoTracker Green (50 nM) for 30 min at 30 °C. After washing, DAPI was added for 20 min at 30 °C, and the ZKM were then washed again and allowed to air dry. Next, the slides were mounted, and three different fields were visualized under a confocal microscope (×100, Nikon Ti2). The quantitative analyzes of the confocal images were performed for two parameters: % ZKM showing fragmented mitochondria, and aspect ratio (AR). The AR was calculated using the ImageJ software.

### 2.17. IL-1β Assay

IL-1β was quantified using the IL-1β ELISA kit (Sunlong Biotech, Hangzhou, China). ZKM (2 × 10^6^) pre-treated with or without inhibitors, or transfected with sc-siRNA or *il1b*-siRNA were infected with *A. hydrophila*. The cultures were terminated at indicated time points p.i., and 100 µL of cell-free culture supernatant was collected and loaded onto IL-1β antibody-precoated wells. The wells were then sealed and incubated at 37 °C for 30 min. Following washing with wash buffer for 3 times, 50 µL of HRP-Conjugate reagent was added to each well; samples were incubated at 37 °C for 30 min and washed as mentioned above. A total of 50 μL of chromogen Solution A and 50 μL of chromogen Solution B were added to each well, mixed, and incubated at 37 °C for 15 min. The absorbance was recorded at A_450_ using a microtiter plate reader immediately after adding 50 μL of stop solution. The concentrations of IL-1β were determined through interpolating the values from the standard curve.

### 2.18. Caspase-1 and Caspase-3 Assays

ZKM (2 × 10^6^) pre-treated with or without inhibitors or transfected with sc-siRNA or *il1b*-siRNAs were infected with *A. hydrophila*. ZKM were washed and lyzed (at 12 h p.i. for caspase-1 and at 24 h p.i. for caspase-3, respectively), and 50 μL of cell lysate was mixed with a 2× reaction buffer containing 5 μL of YVAD-pNA (caspase-1 specific substrate) and DEVD-pNA substrate (caspase-3 specific substrate). The samples were incubated for 1 h at 37 °C and the absorbance was recorded at A_405_ using a microtiter plate reader (Epoch2, BioTek). The relative fold changes in caspase-1 and caspase-3 activities were plotted as compared to the control.

### 2.19. Statistical Analysis

The statistical analyzes were performed with the one-way analysis of variance (ANOVA), followed by Dunnett’s post hoc analysis using GraphPad software. The value of *p* < 0.05 was considered statistically significant for all conducted analyzes.

## 3. Results

### 3.1. A. hydrophila Induces ZKM Apoptosis

*A. hydrophila*-induced cell death has been reported in several organisms including fish [7,42]. To confirm this, ZKM were infected with different MOIs of *A. hydrophila* (ZKM: bacteria), and cytotoxicity was studied using the trypan blue dye exclusion method at 24 h p.i. We observed a positive correlation between ZKM cytotoxicity and MOI (Figure 1). The infected ZKM demonstrated characteristic changes including increased granularity and shrinkage, aggregation, cell rounding, and detachment from the cell culture plates unlike the uniform monolayer with an elongated morphology of uninfected ZKM (data not shown). We selected the MOI 1:50 for subsequent studies as the percentage cytotoxicity was approximately 50%.

The next step was establishing the nature of ZKM death. Towards that direction, ZKM were infected at MOI 1:50, stained with Hoechst 33342, and observed under a fluorescence microscope at 24 h p.i. to detect chromatin condensation, which is characteristic of apoptosis [43]. We noted a significant number of Hoechst-positive ZKM suggestive of apoptosis (Figure 2A). AV binds to exposed phosphatidylserine on the plasma membrane of apoptotic cells while PI stains necrotic cells [44]. *A. hydrophila*-infected ZKM were further stained with AV-PI and observed under a fluorescence microscope at 24 h p.i. We observed a significant number of early apoptotic (AV^+^PI^−^ 52.62 ± 7.67%) and late apoptotic (AV^+^PI^+^, 21.55 ± 5.42%) ZKM, along with an exceptionally few necrotic ZKM (AV^−^PI^+^, 4.38 ± 0.93%) (Figure 2B), thereby establishing *A. hydrophila*-induced ZKM death to be apoptotic. The presence of an oligonucleosomal DNA ladder is a hallmark of apoptosis [45], and we observed the presence of a distinct oligonucleosomal DNA ladder in infected ZKM (Figure 2C). Caspase activation is a key biochemical feature of apoptosis, and the fundamental element of this proteolytic cell death cascade is caspase-3. Hence, we investigated the activity of caspase-3 in *A. hydrophila*-infected ZKM and observed a time-dependent increase in relative caspase-3 activity, with its maximum activity noted at 24 h p.i. (Figure 2D).

Further, pre-treatment of ZKM with the caspase-3 inhibitor, Ac-DEVD-CHO, for 1 h prior to *A. hydrophila* infection led to a significant decline in % Hoechst-positive and AV^+^PI^−^/AV^+^PI^+^ ZKM at 24 h p.i. (Figure 2A,B). Together, these results confirmed that *A. hydrophila*-induced ZKM death is apoptotic in nature, which is mediated by caspase-3.

### 3.2. A. hydrophila Induces Canonical Wnt Signaling in ZKM

Our next step was to identify the signaling molecules that play a role in *A. hydrophila*-induced ZKM apoptosis. Canonical Wnt signaling, reported to regulate inflammatory responses and apoptosis [14,46], was deemed a promising target. ZKM were infected with *A. hydrophila*, and the expression of Wnt2 and Wnt3a, two key ligands for canonical Wnt signaling were monitored by RT-qPCR. We observed significant up-regulations in *wnt2* and *wnt3a* mRNA expression, with peak fold changes recorded at 2 h p.i. for *wnt2* and at 1 h p.i. for *wnt3a*, respectively (Figure 3A). Concordantly, we also observed a significant upregulation in the expression of the Wnt receptor *fzd5* and the co-receptor *lrp6* in these infected ZKM (Figure 3B).

Up-regulated *wnt2*, *wnt3a*, *fzd5, and lrp6* expression levels encouraged us to study β-catenin in *A. hydrophila*-infected ZKM. First, we monitored β-catenin (*ctnnb1*) mRNA expression and observed a significant up-regulation in *ctnnb1* expression with the peak fold change recorded at 6 h p.i. (Figure 3C). We followed this through studying the changes in total β-catenin levels using a specific ELISA kit and observed maximum β-catenin levels at 6 h p.i. (Figure 3D). β-catenin translocates to the nucleus for its effector function [47]. Hence, we measured the nuclear β-catenin in infected ZKM and observed maximum β-catenin levels at 6 h p.i. (Figure 3D). Pre-treatment with the β-catenin specific inhibitors, JW67 and MSAB, led to a decrease in nuclear β-catenin protein levels at 6 h p.i. (Appendix A). On the other hand, pre-treatment with the canonical Wnt signaling activators, LiCl and Laduviglusib, led to increases in nuclear β-catenin protein levels at 6 h p.i. (Appendix A). GSK3β and axin are known to negatively regulate cellular β-catenin levels [47]. In line with this, we monitored *gsk3b* and *axin* mRNA expression, and our results suggested down-regulations in *gsk3b* and *axin* mRNA expression in *A. hydrophila*-infected ZKM (Figure 3E). Collectively, our results suggest that *A. hydrophila* activates canonical Wnt signaling in ZKM.

### 3.3. β-Catenin Mediates NOX-Dependent Oxidative Stress in A. hydrophila-Infected ZKM

After establishing the involvement of β-catenin in *A. hydrophila* infection, we sought to identify the downstream molecules mediating its effects. Oxidative stress, being an integral component of the canonical Wnt signaling pathway [20,30], was deemed an attractive candidate. Hence, ZKM were infected with *A. hydrophila,* and superoxide generation was studied. As we observed a significant increase in superoxide levels with maximum production recorded at 6 h p.i., we thereby selected this time point for subsequent studies (Appendix A). Previous studies have suggested that β-catenin induces ROS and vice versa [20,21]. To study this, ZKM pre-treated with JW67 for 1 h were infected with *A. hydrophila* and superoxide production was monitored at 6 h p.i. The significant reduction in superoxide levels observed suggested that β-catenin positively impacts superoxide generation in *A. hydrophila*-infected ZKM (Figure 4A). The NOX inhibitors Apo and DPI were found to inhibit superoxide generation in the infected ZKM at 6 h p.i. further suggesting β-catenin-mediated superoxide production to be NOX-dependent (Figure 4A). However, we did not notice any effects of Apo or DPI on the nuclear translocation of β-catenin (data not shown). Moreover, we were interested to understand how β-catenin influences ROS generation. Previous studies have suggested that Src homology 2 domain-containing transforming protein 2 (Shc2) plays an important role in regulating ROS generation [48]. To investigate this, we measured *shc2* mRNA expression in the infected ZKM, and observed a maximum fold increase at 6 h p.i. (Appendix A). In the following step, ZKM were pre-treated with JW67 for 1 h, and *shc2* mRNA expression was measured at 6 h p.i., which revealed that *shc2* mRNA expression was significantly repressed in the presence of JW67 at 6 h p.i (Figure 4B) suggesting β-catenin-induced *shc2* expression to be a critical event in augmenting ROS production in *A. hydrophila-*infected ZKM.

### 3.4. β-Catenin-Mediated Cytosolic ROS Positively Impacts mtROS Generation

The role of mtROS in regulating *A. hydrophila* pathogenesis has been well established [35]. The oxidant-rich environment in the infected ZKM prompted us to explore the role of β-catenin in mtROS production. At the outset, ZKM were infected with *A. hydrophila*, and the changes in the mtROS levels were measured using a fluorimeter. We observed enhanced mtROS production with a maximum relative increase recorded at 12 h p.i. (Appendix A). The results were further validated using fluorescence microscopy (Appendix A). Based on these observations, we selected 12 h p.i. for further studies.

In the next step, ZKM pre-treated with JW67 and MSAB for 1h were infected with *A. hydrophila*, and changes in mtROS levels were assessed with fluorimetry and fluorescence microscopy at 12 h p.i. (Figure 5A,B). The significant down-regulation in the mtROS levels implicates the role of β-catenin in triggering mtROS production in *A. hydrophila* infection. In addition, the incubation of the ZKM with canonical Wnt/β-catenin pathway activators (LiCl and Laduviglusib) for 1 h led to increases in mtROS production in the ZKM at 12 h p.i. (Figure 5A,B). The cell-permeable mtROS inhibitor YCG063 and the ETC inhibitor Ant A were used as the negative and positive controls, respectively, for this study (Figure 5A,B). Mitochondrial Sod or Sod2 help in quenching mtROS and maintaining the mitochondrial redox homeostasis. We noted that *sod2* mRNA levels were repressed in the infected ZKM (Appendix A) and were found to be significantly increased in JW67-pretreated ZKM at 12 h p.i. (Figure 5C) suggesting that β-catenin inhibits Sod2 machinery augmenting mtROS production in *A. hydrophila-*infected ZKM.

There are reports suggesting that cellular ROS impacts mtROS generation [29], although the molecular underpinnings are poorly defined. We observed that mtROS production was downregulated in Apo and DPI pre-treated ZKM at 12 h p.i. (Figure 5A,B). To this, we conclude that β-catenin-mediated cellular ROS (NOX-dependent) augments mtROS production, perpetuating the pro-oxidant milieu in *A. hydrophila* infection.

### 3.5. β-Catenin Alters ΔΨ_m_ in A. hydrophila-Infected ZKM

Prolonged mtROS generation disrupts ΔΨ_m_, thereby affecting its functioning [49]. We observed a significant reduction in ΔΨ_m_ in the infected ZKM with a maximum decrease recorded at 12 h p.i., and selected this time point for subsequent studies (Appendix A). In the next step, ZKM pre-treated with JW67 for 1 h were infected, and changes in ΔΨ_m_ were monitored at 12 h p.i. We observed a significant recovery of ΔΨ_m_ in the infected ZKM at 12 h p.i. (Figure 6A). CsA was used as the negative control for this study (Figure 6A). Together, our results indicate that β-catenin-induced mtROS reduces the ΔΨ_m_ in *A. hydrophila*-infected ZKM.

### 3.6. β-Catenin-Induced mtROS Abets Mitochondrial Fission in A. hydrophila-Infected ZKM

Elevated mtROS levels alter mitochondrial dynamics (namely fission and fusion) [50]. To explore this, we monitored the expression of Drp1 (*dnm1l*) in infected ZKM. Drp1 favors mitochondrial fission, and we observed a significant increase in *dnm1l* mRNA expression with the maximum fold change observed at 12 h p.i. (Appendix A). Additionally, the expression of genes favoring mitochondrial fusion was also studied, and we noted that the expression of both *mfn1* and *mfn2* were significantly down-regulated in the infected ZKM (Appendix A). Our findings indicate a shift in mitochondrial dynamics towards fission in the infected ZKM. *A. hydrophila*-induced mitochondrial fission was further confirmed with confocal microscopy (Figure 6E). The uninfected ZKM showed regular, uniform distributions of the elongated mitochondria which subsequently altered into short, fragmented segments in *A. hydrophila*-infected ZKM (Figure 6E).

The next step was studying the role of β-catenin in regulating the mitochondrial dynamics. Thus, ZKM pre-treated with JW67 and MSAB for 1 h were infected with *A. hydrophila,* and the expression of *dnm1l*, *mfn1*, and *mfn2* mRNA were monitored at 12 h p.i. Pre-treatment with JW67 and MSAB were found to downregulate *dnm1l* expression and upregulate the expression of *mfn1* and *mfn2* at 12 h p.i. (Figure 6B–D). The same was noted using the Drp1 inhibitor Mdivi-1 and the mtROS inhibitor YCG063, respectively (Figure 6B–D). We also observed that pre-treatment with JW67, MSAB, Mdivi-1 and YCG063 for 1 h led to the rescue of the mitochondrial network in the infected ZKM at 12 h p.i. (Figure 6C). In addition, ZKM incubated with LiCl and Laduviglusib for 1 h showed increases in *dnm1l* expression, and decreases in *mfn1* and *mfn2* expression, along with extensive mitochondrial fragmentation at 12 h p.i. (Figure 6B–E). We then analyzed the mitochondrial network through determining the aspect ratio estimating the network elongation and % ZKM showing fragmented mitochondria. These analyzes revealed elongation of the mitochondrial fragments, detected at 12 h p.i. and a decrease in % ZKM showing mitochondrial fission in the presence of the β-catenin inhibitors (JW67 and MSAB), the mtROS inhibitor (YCG063), and the Drp1 inhibitor (Mdivi-1), while the opposite was observed in presence of the mtROS inducer (Ant A) and the Wnt/β-catenin pathway activators (LiCl and Laduviglusib). Based on these findings, we suggest that β-catenin-induced mtROS facilitates Drp1-mediated mitochondrial fission in *A. hydrophila*-infected ZKM.

### 3.7. β-Catenin-Induced Mitochondrial Fission Favors ZKM Apoptosis and Pathogen Removal

The involvement of β-catenin in *A. hydrophila* infection made us interested to study its role in pathogenesis. First, ZKM pre-treated with JW67 for 1 h were infected with *A. hydrophila,* and then apoptosis was assessed for at 24 h p.i. The significant reduction in the number of Hoechst-positive ZKM observed, as well as in the AV^+^PI^-^ and AV^+^PI^+^ ZKM in the presence of JW67 and MSAB, along with the significant increase in the number of Hoechst-positive ZKM as well as the AV^+^PI^-^ and AV^+^PI^+^ ZKM in the presence of LiCl and Laduviglusib at 24 h p.i. established the pro-apoptotic role of β-catenin in *A. hydrophila* pathogenesis (Figure 7A,B). In the next step, ZKM pre-treated with JW67 for 1 h were infected with *A. hydrophila,* and the intracellular bacteria were enumerated at 24 h p.i. We observed an increased intracellular bacterial load in JW67-pre-treated ZKM, thereby implicating the bactericidal role of β-catenin in *A. hydrophila* infection (Figure 7C).

We questioned the role of β-catenin-induced mitochondrial fission in triggering ZKM apoptosis. Mitochondrial fission is intimately associated with the release of pro-apoptotic cyt *c* into the cytoplasm [51]. Hence, ZKM were infected with *A. hydrophila,* and the cytosolic cyt *c* levels were measured. We recorded the maximum accretion of cyt *c* in the cytoplasm at 12 h p.i. and selected this time interval for subsequent studies (Appendix A). In the next step, ZKM were pre-treated with JW67 and Mdivi-1 for 1 h, respectively, and the changes in cytosolic cyt *c* levels measured at 12 h p.i. We observed significantly reduced cyt *c* accumulation, thereby implicating that β-catenin-induced mitochondrial fission triggers cyt *c* release in *A. hydrophila*-infected ZKM (Figure 8A). Cyclosporin A (CsA) prevents the release of cyt *c*, thereby inhibiting apoptosis (Walter et al., 1998). Pre-treatment with CsA for 1 h inhibited cyt *c* release at 12 h p.i. (Figure 8A), and attenuated ZKM apoptosis at 24 h p.i. (Figure 7A).

We concluded this study by investigating the role of mitochondrial fission on bacterial growth. ZKM pre-treated with Mdivi-1 were infected with *A. hydrophila* and intracellular bacterial growth was studied at 24 h p.i. We observed that blocking mitochondrial fission led to an increase in intracellular replication of *A. hydrophila* (Figure 7C). Our results, for the first time, indicated the pro-apoptotic and bactericidal role of the β-catenin-Drp1 axis in *A. hydrophila*-infected ZKM.

### 3.8. Cyt c Induces Pro-Apoptotic Caspase-1/IL-1β/Caspase-3 Axis in A. hydrophila-Infected ZKM

Among several downstream signaling events activated in *A. hydrophila* infection, caspase-1/IL-1β is important [7]. On the outset, we monitored caspase-1 and IL-1β in the infected ZKM and observed maximum caspase-1 activity and IL-1β levels at 12 h p.i. and selected this time interval for subsequent studies (Appendix A). In the next step, ZKM pre-treated with Mdivi-1 and CsA for 1 h were infected, and the changes in caspase-1 activity and IL-1β levels were monitored at 12 h p.i. We observed attenuated caspase-1 activity and a concomitant decline in IL-1β levels at 12 h p.i. in Mdivi-1 and CsA pre-treated ZKM (Figure 8B,C). The caspase-1 inhibitor (Z-YVAD-FMK) and *il1b*-siRNA were used as the respective controls for the study. Additionally, we noted significant caspase-3 activity in *A. hydrophila*-infected ZKM (Appendix A), and pre-treatment with Z-YVAD-FMK and *il1b*-siRNA, respectively, resulted in a marked inhibition in caspase-3 activity at 24 h p.i. (Figure 8D), along with a concomitant decline in ZKM apoptosis at 24 h p.i. (Figure 7A). These results suggest the presence of the inductive effect of the β-catenin/Drp1/cyt *c* axis on caspase-1/IL-1β activation to induce caspase-3-mediated apoptosis of *A. hydrophila*-infected ZKM.

## 4. Discussion

In the present study, we have established that canonical Wnt signaling acts as a pro-apoptotic bactericidal signalosome, which functions as an innate immune attribute in resolving *A. hydrophila* infection.

The dose of infectious agent prosecutes host cell death [52]. We observed *A. hydrophila*-induced ZKM death to be MOI-dependent, suggesting bacterial load as a major contributor to *A. hydrophila*-induced ZKM cytotoxicity. Our results are in accordance with earlier reports of *A. hydrophila*-induced MOI-dependent death of macrophages [53]. Bacteria-induced host cell death is an intrinsic immune defense mechanism [54]. Two major cell death modalities triggered following infection are necrosis and apoptosis, and the activation of both pathways have been reported in *A. hydrophila-*infected fish macrophages [34,55]. We noticed that *A. hydrophila* induces caspase-mediated ZKM apoptosis, which prompted us to identify the upstream signaling molecules involved in this process.

The role of canonical Wnt signaling in regulating innate immune pathways has not been well understood in fish. Upregulated expression of Wnt machinery coupled with the decreased expression of the β-catenin destruction complex components suggested the activation of canonical Wnt signaling was consequent to *A. hydrophila* infection in ZKM. A transcriptome-based approach previously suggested the involvement of canonical Wnt signaling in zebrafish upon infection with *A. hydrophila* [9]. The suppressed expression of the β-catenin destruction complex components is indicative of nuclear β-catenin accumulation, thereby inducing the expression of β-catenin-dependent genes in ZKM. This led us to investigate the subcellular distribution of β-catenin, and enhanced β-catenin levels were indeed observed in the nucleus of the infected ZKM, thereby indicating its proactive role in *A. hydrophila* infection. Our results are in line with studies reporting bacteria-induced nuclear translocation of β-catenin thereby influencing pathogenesis [56,57].

Previous studies traversing the importance of canonical Wnt signaling in cellular homeostasis have suggested its pro-and anti-apoptotic effects in bacterial pathogenesis [14,46,58]. We sought to correlate canonical Wnt signaling with ZKM apoptosis and observed that inhibiting β-catenin activity repressed ZKM apoptosis. This is the first report suggesting a pro-apoptotic role of β-catenin in *A. hydrophila* infection. The next step was to understand how β-catenin induced ZKM apoptosis. Among the several downstream targets of β-catenin, ROS is particularly important [20,30]. The role of ROS in restraining *A. hydrophila* has been well reported in fish [34]. ROS is produced from diverse sources of which the NOX-pathway is important [59]. We observed a significant upregulation in the NOX-induced superoxide levels, which was previously suppressed in the presence of the β-catenin inhibitor. Additionally, suppressing the NOX-pathway led to concomitant increases in intracellular *A. hydrophila* replication. Based on these results, we suggest that β-catenin enhances NOX-induced ROS production, thereby triggering ZKM apoptosis and *A. hydrophila* clearance. β-catenin-mediated ROS production with cytotoxic implications has been reported previously [60,61]. However, we do not know the potential mechanisms for β-catenin-induced ROS production. Several NOX-isoforms are present in fish, and identifying the particular isoform that is regulated by β-catenin, along with examining how β-catenin induces ROS will help in understanding the molecular underpinnings of *A. hydrophila* pathogenesis.

Shc2 is an adaptor protein which promotes ROS production and regulates cellular responses to oxidative stress [48]. From observing the elevated ROS levels in infected ZKM, we hypothesized that β-catenin impacts Shc2 activity. To assess this, we monitored *shc2* mRNA expression in the presence and absence of β-catenin signaling. The observed β-catenin-induced increase in *shc2* mRNA expression suggests that Shc2 acts as a transducing molecule in the β-catenin-ROS axis in *A. hydrophila-*infected ZKM. However, how β-catenin influences Shc2 functioning is still not clear, and therefore merits future investigations.

Mitochondria are another major source of cellular ROS (mtROS). Although mtROS generation in infected macrophages has been reported both in vitro and in vivo, the underlying molecular mechanisms remain poorly defined. Moreover, reports investigating ROS-induced-mtROS production are limited in fish. We observed significant mtROS levels in *A. hydrophila-*infected ZKM. Inhibiting mtROS resulted in increased intracellular bacterial load, thereby implicating its antibacterial role in *A. hydrophila* pathogenesis. This is in line with previous studies reporting mtROS production as consequent to infection by *A. hydrophila* [62] and other microbes in fish [63], establishing mtROS generation to be a conserved innate immune attribute in fish. Among several cues triggering mtROS generation, NOX-induced ROS is particularly important [64]. We noted that inhibiting NOX-induced ROS production interfered with mtROS generation. To the best of our knowledge, this is the first report on ROS-induced mtROS production in fish. Contrary to previous studies documenting mtROS prompts canonical Wnt signaling [65], we observed that β-catenin triggers mtROS production in *A. hydrophila-*infected ZKM. To this, we conclude that the positive feed-forward mechanisms of β-catenin and NOX promoting mtROS production is a self-amplifying process for ROS accumulation in *A. hydrophila*-infected ZKM. Our results may help in explaining the pro-oxidant role of the β-catenin/NOX axis in ZKM apoptosis and pathogenesis of *A. hydrophila* in fish.

Sod2 plays an important role in maintaining the redox balance under different conditions of stress [66]. Its role has been well reported in bacterial infections in fish [67]. We observed compromised *sod2* mRNA expression in *A. hydrophila*-infected ZKM, which recovered on inhibiting β-catenin signaling. Based on these results, we propose that β-catenin inhibits the activation of cellular antioxidant machinery leading to pronounced oxidative stress (NOX-mediated and mtROS), thereby eliciting ZKM apoptosis and *A. hydrophila* clearance.

Synchronism in mtROS production and ΔΨ_m_ collapse has also been well documented [29,49], but the link between these two molecular events has not been well understood in *A. hydrophila* pathogenesis. We report that mtROS induces the decrease of ΔΨ_m_ in *A. hydrophila*-infected ZKM. Mitochondrial dynamics plays an intimate role in apoptosis, and it has been suggested that the loss in ΔΨ_m_ favors mitochondrial fission, triggering apoptosis [68]. We set out to unravel the role of altered ΔΨ_m_ in mitochondrial fission. The key players regulating mitochondrial fission include Drp1, Mfn1, and Mfn2, respectively [69]. We observed that *A. hydrophila* infection results in the upregulation in pro-fission *dnm1l* expression, and downregulation in anti-fission *mfn1* and *mfn2* expression, which was reversed in the presence of CsA and Mdivi-1. Inhibition of MPTP and mitochondrial fission also restored the intracellular growth of *A. hydrophila*. Additionally, inhibiting β-catenin signaling restored ΔΨ_m_, decreased *dnm1l* expression, and increased *mfn1* and *mfn2* expression, with concomitant reductions in mitochondrial fission and increased intracellular *A. hydrophila* replication observed. Collectively, our results implicate that: (1) β-catenin induces mitochondrial fission using mtROS and ΔΨ_m_ as intermediates, and (2) mitochondrial fission leads to apoptosis of the infected ZKM and clearance of intracellular bacteria.

Mitochondrial fission results in the leakage of mitochondrial proteins in the cytosol, activating the downstream apoptotic signaling cascade [70]. We observed mitochondrial fission was intimately related to increased cytosolic cyt *c* accumulation. Our results corroborate previous findings depicting inhibition in mitochondrial fission leads to the reduced release of cyt *c* into the cytosol [71,72]. However, while *A. hydrophila* infection has been reported to induce cyt *c* release subsequently triggering apoptosis [34], the role of β-catenin as an upstream inducer has not been reported as of yet. We affirmed this as inhibition of β-catenin signaling leading to attenuated cytosolic cyt *c* levels in the infected ZKM. Hence, our study reflects on the involvement of β-catenin in inducing mitochondrial fission leading to the release of pro-apoptotic cyt *c* into the cytosol.

The caspase-1/IL-1β signalosome has been implicated in inducing caspase-3-mediated apoptosis in *A. hydrophila* pathogenesis [7]. Extending our previous studies, we report the role of β-catenin in actuating the caspase-1/IL-1β/caspase-3 axis in *A. hydrophila*-infected fish macrophages. These results are in synchrony with earlier studies on canonical Wnt signaling triggering caspase-dependent apoptosis in bacterial infections [58]. Together, our results demonstrate that mitochondrial fission-induced inflammatory milieu contributes to the apoptosis of infected ZKM and bacterial clearance.

## 5. Conclusions

In conclusion, our study has established the pro-apoptotic and bactericidal role of canonical Wnt signaling in *A. hydrophila* pathogenesis. We propose that β-catenin creates a pro-oxidant environment which triggers mitochondrial fission, culminating in caspase 3-mediated apoptosis of the infected ZKM, and the clearance of intracellular *A. hydrophila* (Figure 9). Our findings cement the role of mitochondrial fission in activating inflammatory signalosomes, which impact fish innate immunity, and paves the way for ascertaining potential targets for future therapeutic interventions in controlling the disease spectrum in both fish and humans.

## Figures and Tables

**Figure 1 cells-12-01509-f001:**
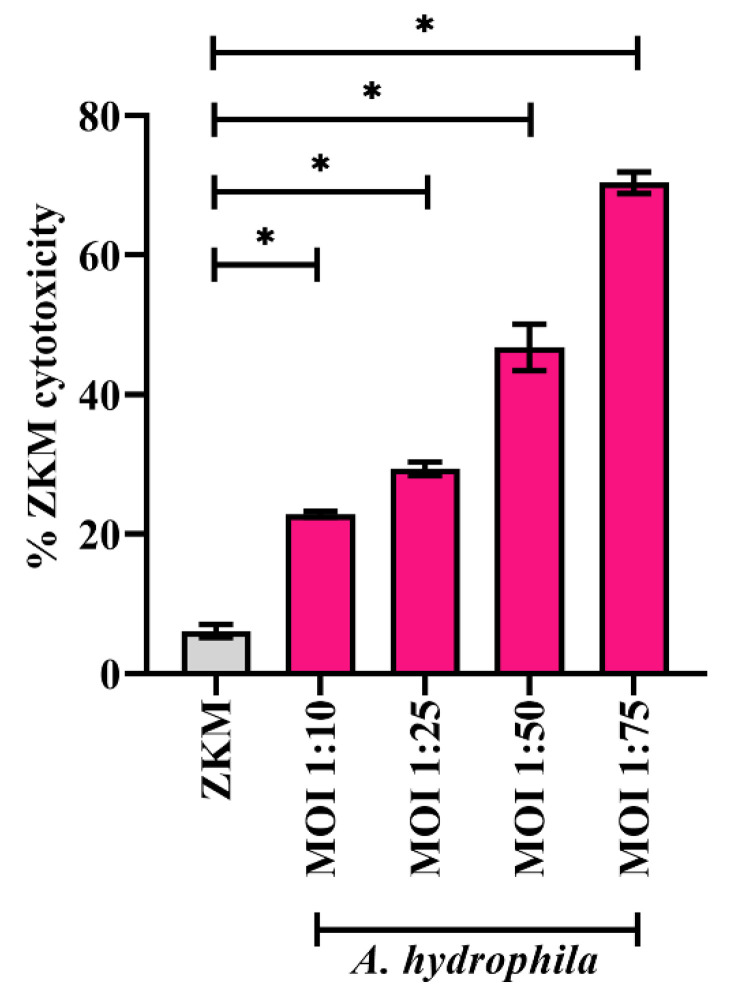
*A. hydrophila*-infected ZKM show MOI-dependent death. ZKM were infected with an indicated MOI (ZKM:bacteria) and % ZKM cytotoxicity was enumerated at 24 h p.i. using 0.4% trypan blue dye exclusion method. At least 100 cells per field were observed for determining the % ZKM cytotoxicity. Vertical bars denote the mean ± SEM (n = 3). Asterisk (*) denotes the significant difference compared to the control (* *p* < 0.05).

**Figure 2 cells-12-01509-f002:**
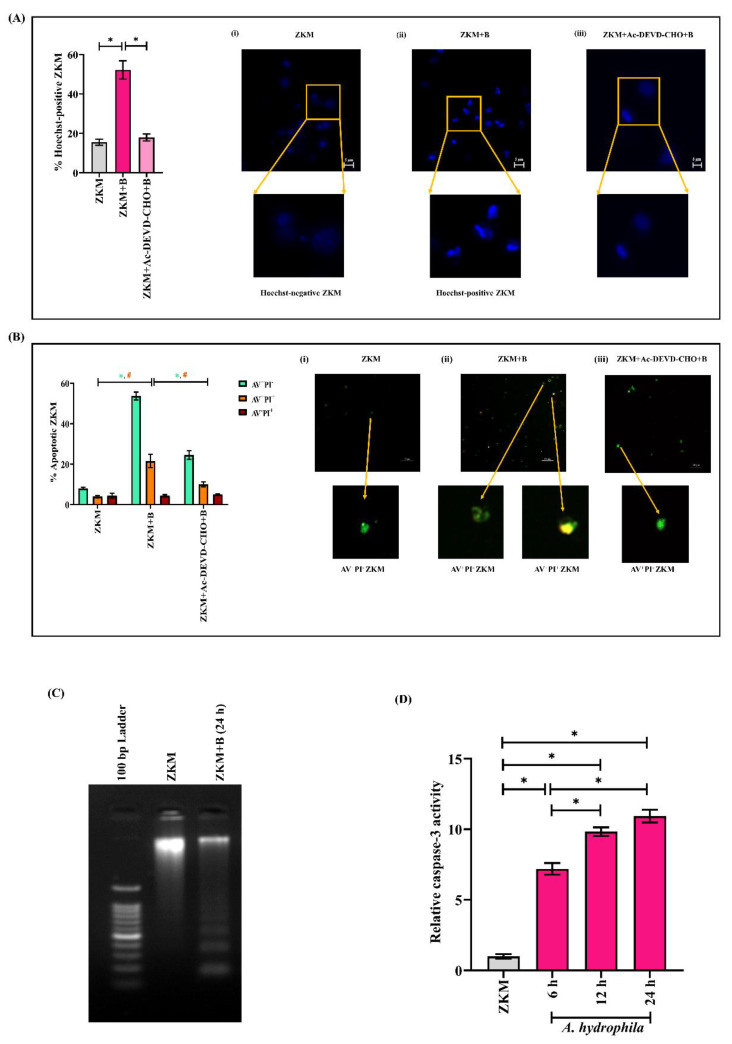
*A. hydrophila*-induced ZKM death is apoptotic in nature. (**A**) ZKM pre-treated with or without the caspase-3 inhibitor (Ac-DEVD-CHO) were infected with *A. hydrophila* (MOI 1:50), and apoptosis was assessed by enumerating Hoechst-positive ZKM at 24 h p.i., A(i), A(ii), and A(iii) represent the uninfected ZKM, *A. hydrophila*-infected ZKM, and *A. hydrophila*-infected ZKM pre-treated with Ac-DEVD-CHO, respectively, at 24 h p.i. (×40). Three different fields having at least 100 cells were observed for determining the percentage of Hoechst-positive ZKM. (**B**) B(i) represents uninfected ZKM, B(ii) *A. hydrophila*-infected ZKM, and B(iii) *A. hydrophila*-infected ZKM pre-treated with the caspase-3 inhibitor (Ac-DEVD-CHO), respectively, were stained with AV-PI at 24 h p.i., and photomicrographs were captured using the fluorescence microscope (×40). Three different fields having at least 100 cells were observed for determining the percentages of AV^+^PI^+^, AV^−^PI^+^, and AV^+^PI^-^ cells. (**C**) Uninfected and *A. hydrophila*-infected ZKM were lysed at 24 h p.i., with their gDNA then isolated and subjected to agarose gel electrophoresis alongside the 100 bp DNA marker. (**D**) ZKM were infected with *A. hydrophila,* and relative caspase-3 activity was measured at indicated time points p.i. Vertical bars denote the mean ± SEM (n = 3). Asterisk (*) and hash (#) denote a significant difference between the indicated groups (* *p* < 0.05). “+B” mentioned in the *X*-axis represents “+*A. hydrophila*”.

**Figure 3 cells-12-01509-f003:**
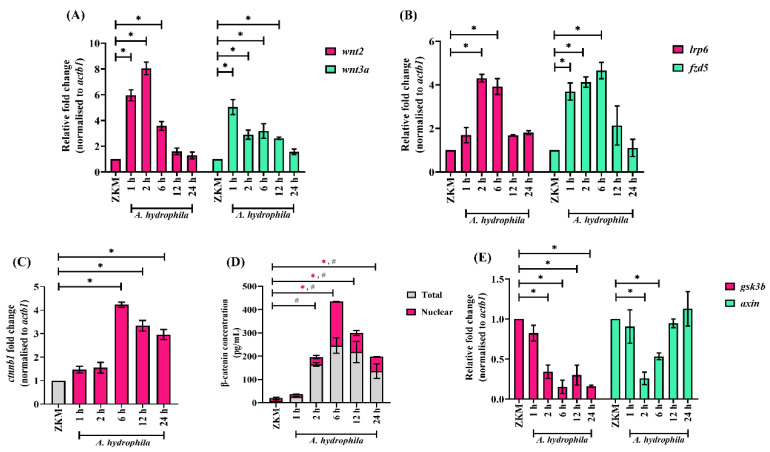
Canonical Wnt signaling is activated in *A. hydrophila*-infected ZKM. ZKM were infected with *A. hydrophila*, and at indicated time points p.i., mRNA expression analysis was conducted for (**A**) *wnt2* and *wnt3a*, (**B**) *lrp6* and *fzd5*, (**C**) *ctnnb1*, and (**E**) *gsk3b* and *axin* using RT-qPCR. (**D**) ZKM were infected with *A. hydrophila*, and at indicated time points p.i., total β-catenin and nuclear β-catenin protein levels were quantified. Vertical bars represent the mean ± SEM (n = 3). Asterisk (*) and hash (#) denote the significant difference compared to the control (* *p* < 0.05).

**Figure 4 cells-12-01509-f004:**
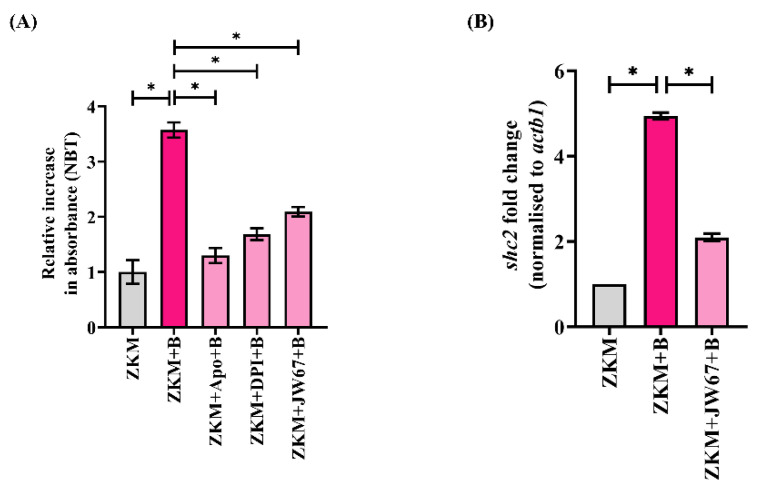
β-catenin triggers NOX-induced ROS production and upregulates *shc2* mRNA expression. (**A**) ZKM pre-treated with or without NOX inhibitors (Apo and DPI) and β-catenin inhibitor (JW67) for 1 h were infected with *A. hydrophila*, and superoxide levels were measured using the NBT assay at 6 h p.i., and (**B**) ZKM pre-treated with or without JW67 for 1 h were infected with *A. hydrophila*, and *shc2* mRNA expression was analyzed with RT-qPCR at 6 h p.i. Vertical bars represent the mean ± SEM (n = 3). Asterisk (*) denotes a significant difference between the indicated groups (* *p* < 0.05). “+B” mentioned in the *X*-axis represents “+*A. hydrophila*”.

**Figure 5 cells-12-01509-f005:**
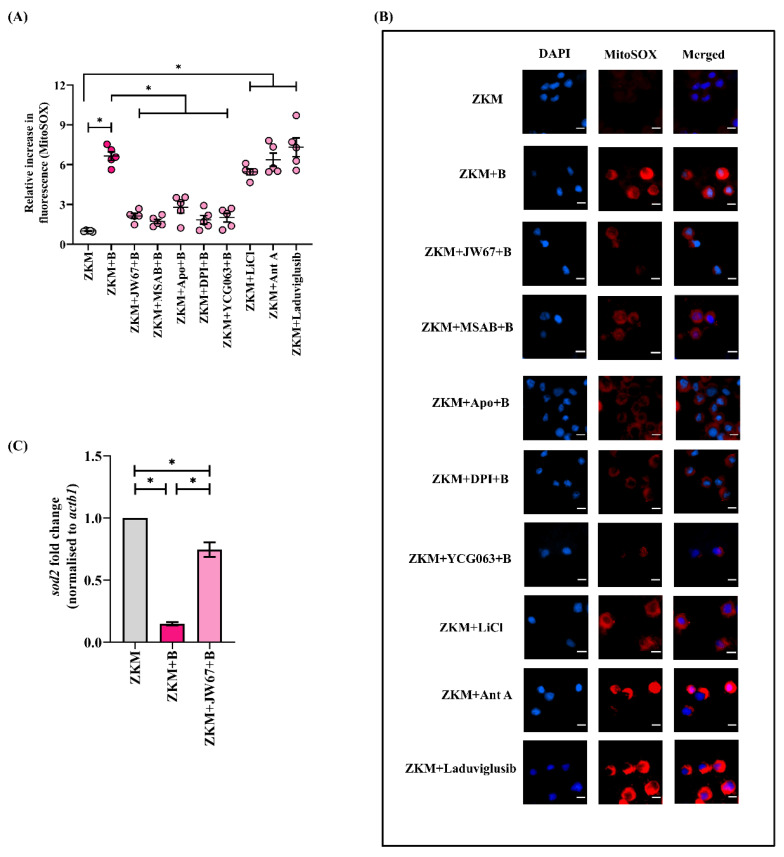
β-catenin triggers ROS-induced-mtROS generation and downregulates *sod2* mRNA expression. ZKM pre-treated with or without β-catenin inhibitors (JW67 and MSAB), NOX inhibitors (Apo and DPI), and mtROS inhibitor (YCG063) for 1 h were infected with *A. hydrophila,* and at 12 h p.i., (**A**) changes in mtROS levels were measured with fluorimetry, and (**B**) changes in mtROS levels were visualized under the fluorescence microscope. Fluorescence microscopic data are representative of three independent experiments. Ant A was used as a positive control. Additionally, ZKM were incubated with canonical Wnt/β-catenin pathway activators (LiCl and Laduviglusib), and changes in mtROS levels were measured with both fluorimetry and fluorescence microscopy. (**C**) ZKM pre-treated with or without the β-catenin inhibitor (JW67) were infected with *A. hydrophila* and *sod2* mRNA expression was analyzed with RT-qPCR at 12 h p.i. Vertical bars denote the mean ± SEM (n = 3). Asterisk (*) denotes a significant difference between the indicated groups (* *p* < 0.05). “+B” mentioned in the *X*-axis and microscopy images represents “+*A. hydrophila*”.

**Figure 6 cells-12-01509-f006:**
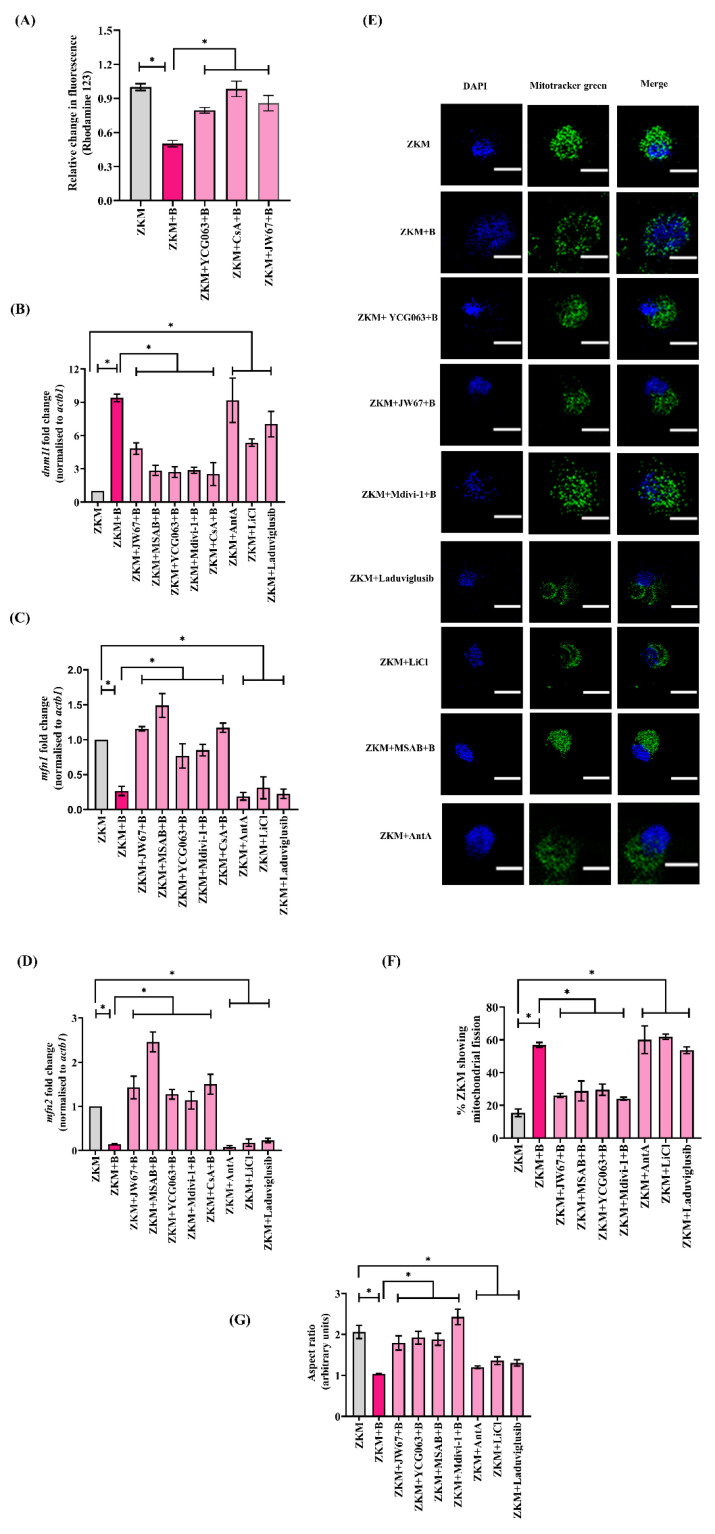
β-catenin-induced mtROS prompts ΔΨ_m_ loss leading to Drp1-mediated mitochondrial fission in *A. hydrophila*-infected ZKM. (**A**) ZKM pre-treated with or without the mtROS inhibitor (YCG063), MPTP inhibitor (CsA), and β-catenin inhibitor (JW67) for 1 h were infected with *A. hydrophila,* and relative changes in the ΔΨ_m_ were recorded at 12 h p.i. using rhodamine 123. (**B**–**D**) ZKM pre-treated with or without the β-catenin inhibitors (JW67 and MSAB), mtROS inhibitor (YCG063), Drp1 inhibitor (Mdivi-1), and MPTP inhibitor (CsA) for 1 h were infected with *A. hydrophila,* and at 12 h p.i., *dnm1l*, *mfn1,* and *mfn2* mRNA expression analysis were performed using RT-qPCR. Additionally, ZKM were incubated with canonical Wnt/β-catenin pathway activators (LiCl and Laduviglusib) and Ant A for 1 h, and at 12 h p.i., *dnm1l*, *mfn1,* and *mfn2* mRNA expression analysis were performed using RT-qPCR. (**E**) ZKM pre-treated with or without the β-catenin inhibitors (JW67 and MSAB), mtROS inhibitor (YCG063), and Drp1 inhibitor (Mdivi-1) for 1 h were infected with *A. hydrophila,* and the morphology of the mitochondrial network was examined at 12 h p.i. ZKM were washed, stained with MitoTracker green and DAPI, mounted and visualized under microscope (scale-5 µm). Similarly, ZKM were incubated with canonical Wnt/β-catenin pathway activators (LiCl and Laduviglusib) and Ant A and at 12 h p.i., and the morphology of the mitochondrial network was studied at 12 h p.i. Fluorescence microscopic data are representative of three independent experiments. (**F**) Quantification of the percentage of ZKM displaying fragmented mitochondria at 12 h p.i. with or without the pre-treatment of the indicated inhibitors and agonists. Data represents the cumulative result of three independent analyses. (**G**) Quantitative analysis of the aspect ratio in ZKM with or without the pre-treatment of the indicated inhibitors and agonists at 12 h p.i. Data represents the cumulative results of three independent analyses (20 ZKM per experiment). Vertical bars denote the mean ± SEM (n = 3). Asterisk (*) denotes a significant difference between the indicated groups (* *p* < 0.05). “+B” mentioned in the *X*-axis and microscopy images represents “+*A. hydrophila*”.

**Figure 7 cells-12-01509-f007:**
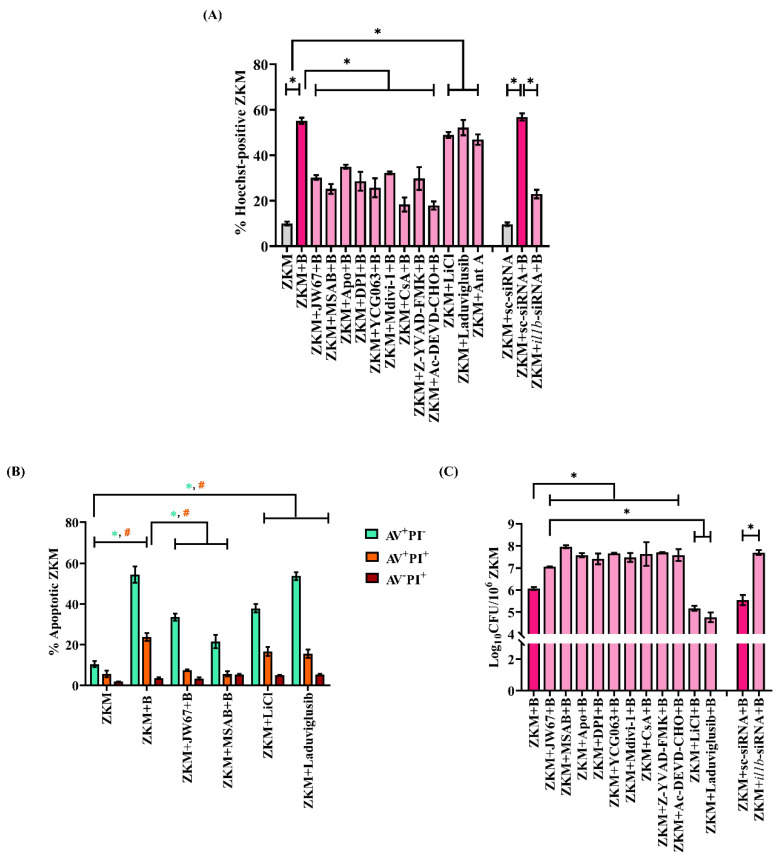
β-catenin plays a pro-apoptotic and anti-bacterial role in *A. hydrophila*-infected ZKM. (**A**) ZKM pre-treated with or without β-catenin inhibitors (JW67 and MSAB), NOX inhibitors (Apo and DPI), mtROS inhibitor (YCG063), Drp1 inhibitor (Mdivi-1), MPTP inhibitor (CsA), caspase-1 inhibitor (Z-YVAD-FMK), and caspase-3 inhibitor (Ac-DEVD-CHO) for 1 h or transfected with sc-siRNA, *il1b*-siRNA were infected with *A. hydrophila,* and % Hoechst-positive ZKM were enumerated at 24 h p.i. Additionally, ZKM were incubated with canonical Wnt/β-catenin pathway activators (LiCl and Laduviglusib) and Ant A for 1 h and % Hoechst-positive ZKM were enumerated at 24 h p.i. (**B**) ZKM pre-treated with or without β-catenin inhibitors (JW67 and MSAB) for 1 h were infected with *A. hydrophila* and stained with AV-PI at 24 h p.i. Next, % apoptotic ZKM were calculated. Similarly, ZKM were incubated with canonical Wnt/β-catenin pathway activators (LiCl and Laduviglusib) for 1 h and % Hoechst-positive ZKM were enumerated at 24 h p.i. (**C**) ZKMs pre-treated with or without β-catenin inhibitors (JW67 and MSAB), NOX inhibitors (Apo and DPI), mtROS inhibitor (YCG063), Drp1 inhibitor (Mdivi-1), MPTP inhibitor (CsA), caspase-1 inhibitor (Z-YVAD-FMK), and caspase-3 inhibitor (Ac-DEVD-CHO) for 1 h or transfected with sc-siRNA, *il1b*-siRNA were infected with *A. hydrophila* and the bacterial loads were enumerated at 24 h p.i. Additionally, ZKM were incubated with canonical Wnt/β-catenin pathway activators (LiCl and Laduviglusib) and Ant A for 1 h and % Hoechst-positive ZKM were enumerated at 24 h p.i. Vertical bars represent the mean ± SEM (n = 3). Asterisk (*) and hash (#) denote a significant difference between the indicated groups (* *p* < 0.05). “+B” mentioned in the *X*-axis represents “+*A. hydrophila*”.

**Figure 8 cells-12-01509-f008:**
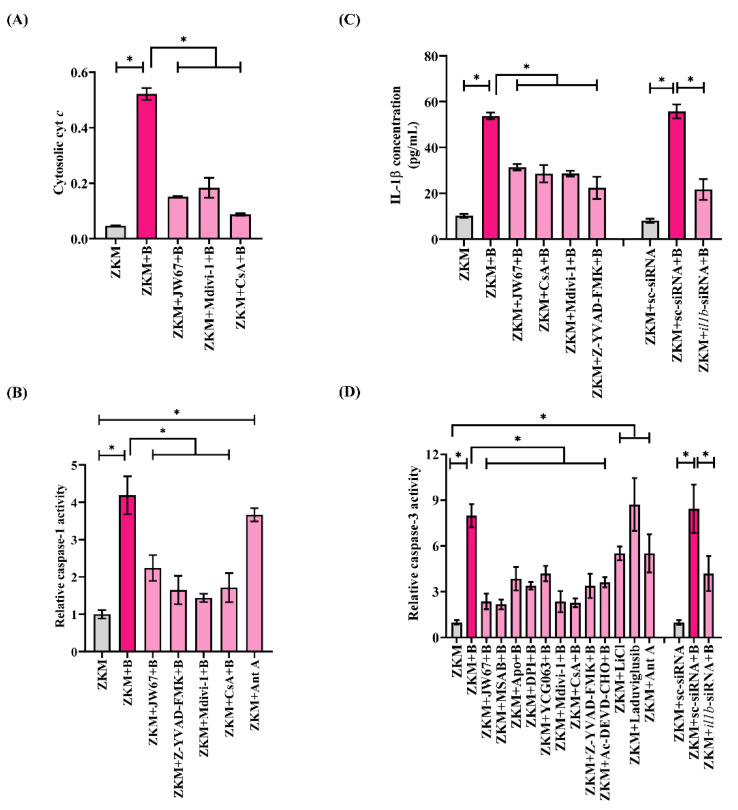
β-catenin-induced mitochondrial fission triggers cyt *c* release, activating the caspase-1/IL-1β/caspase-3 axis in *A. hydrophila*-infected ZKM. (**A**) ZKM pre-treated with or without β-catenin inhibitor (JW67), Drp1 inhibitor (Mdivi-1), and MPTP inhibitor (CsA) for 1 h were infected with *A. hydrophila,* and changes in cytosolic cyt *c* were studied at 12 h p.i. (**B**) ZKM pre-treated with β-catenin inhibitor (JW67), caspase-1 inhibitor (Z-YVAD-FMK), Drp1 inhibitor (Mdivi-1), and MPTP inhibitor (CsA) for 1 h were infected with *A. hydrophila,* and relative changes in caspase-1 activity were studied at 12 h p.i. mtROS inducer (Ant A) was used as a positive control in this study. (**C**) ZKM pre-treated with β-catenin inhibitor (JW67), MPTP inhibitor (CsA), Drp1 inhibitor (Mdivi-1), and caspase-1 inhibitor (Z-YVAD-FMK) for 1 h or transfected with sc-siRNA, *il1b*-siRNA were infected with *A. hydrophila,* and relative changes in IL-1β concentration were plotted at 12 h p.i. (**D**) ZKM pre-treated with or without β-catenin inhibitors (JW67 and MSAB), NOX inhibitors (Apo and DPI), mtROS inhibitor (YCG063), Drp1 inhibitor (Mdivi-1), MPTP inhibitor (CsA), caspase-1 inhibitor (Z-YVAD-FMK), and caspase-3 inhibitor (Ac-DEVD-CHO) for 1 h or transfected with sc-siRNA, *il1b*-siRNA were infected with *A. hydrophila,* and relative changes in caspase-3 activity were studied at 24 h p.i. Additionally, ZKM were incubated with canonical Wnt/β-catenin pathway activators (LiCl and Laduviglusib) and Ant A, and relative changes in caspase-3 activity were studied at 24 h p.i. Vertical bars denote the mean ± SEM (n = 3). Asterisk (*) denotes a significant difference between the indicated groups (* *p* < 0.05). “+B” mentioned in the *X*-axis represents “+*A. hydrophila*”.

**Figure 9 cells-12-01509-f009:**
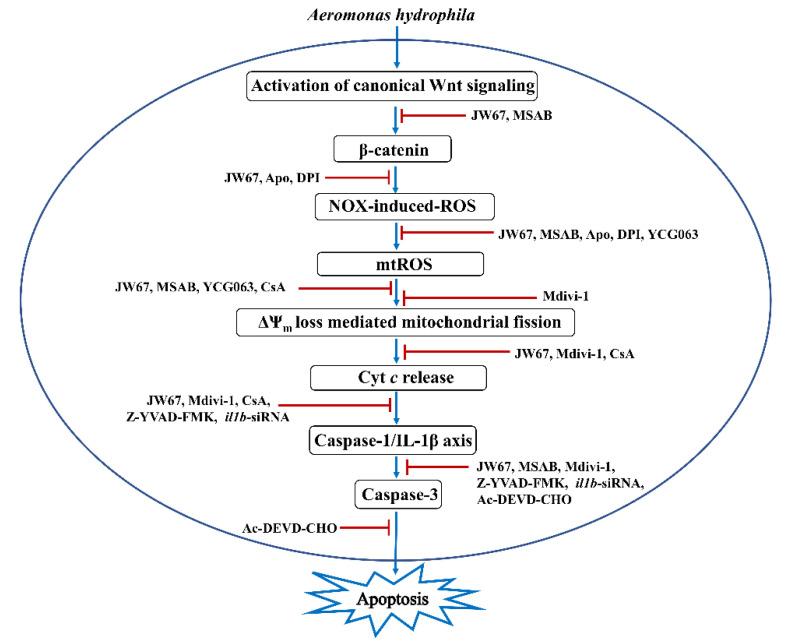
Overview of the study. Canonical Wnt signaling-induced NOX-mediated ROS triggers mtROS generation leading to downstream ΔΨ_m_ loss. ΔΨ_m_ loss prompts the activation of Drp1-mediated mitochondrial fission. Activated mitochondrial fission leads to cyt *c* release, which activates the caspase-1/IL-1β/caspase-3 axis, leading to apoptosis of the *A. hydrophila*-infected ZKM.

**Table 1 cells-12-01509-t001:** List of chemicals used in the study.

S. No.	Chemicals Used	Company	Function
1.	JW67	Sigma	β-catenin inhibitor
2.	MSAB	MedChemExpress	β-catenin inhibitor
3.	LiCl	Sigma	Wnt/β-catenin signaling activator
4.	Laduviglusib	MedChemExpress	Wnt/β-catenin signaling activator
5.	Apo	Sigma	NADPH oxidase (NOX) inhibitor
6.	DPI	Sigma	NADPH oxidase (NOX) inhibitor
7.	Mdivi-1	Sigma	Drp1 inhibitor
8.	CsA	Sigma	Mitochondrial permeability transition pore (MPTP) inhibitor
9.	YCG063	Calbiochem	Mitochondrial ROS (mtROS) inhibitor
10.	Ant A	Sigma	Mitochondrial ROS (mtROS) inducer
11.	Z-YVAD-FMK	Sigma	Caspase-1 inhibitor
12.	Ac-DEVD-CHO	Biovision	Caspase-3 inhibitor

**Table 2 cells-12-01509-t002:** List of primers used for RT-qPCR analysis.

S. No.	Gene Name	Accession Number	Primer Sequence	Amplicon Length (bp)
1.	*actb1*	FJ915059.1	FP:5′-CGAGCAGGAGATGGGAACC-3′ RP: 5′-CAACGGAAACGCTCATTGC-3′	104
2.	*wnt2*	NM_130950.1	FP:5′-CACAATCTGTTCGGGAGGCT-3′ RP: 5′-ACGCCCTGGTCAATGTGTAG-3′	100
3.	*wnt3a*	NM_001007185.1	FP:5′-TAAGCAAGCAAAGGCCACCAG-3′ RP: 5′-GACACCATGCTGCCGAACTC-3′	198
4.	*lrp6*	NM_001134684.1	FP:5′-GTCAACACACCGCTCCTACA-3′ RP:5′-CCCGGCGTATAGTCACTGTC-3′	105
5.	*fzd5*	NM_131134.2	FP:5′-CCTTGCCACCAACCCTACTT-3′ RP: 5′-CGCTCCATGTCGATGAGGAA-3′	128
6.	*gsk3b*	NM_131381.1	FP:5′-CATCTTTGGAGCCACCGACT-3′ RP: 5′-TGGCCGAAACACCTTAGTCC-3′	240
7.	*axin*	NM_131503.2	FP: 5′-TCTGGCCAATCACAGGGTTC-3′ RP: 5′-TCGTGTGCATCCCTTAGCTG-3′	102
8.	*ctnnb1*	NM_131059.2	FP:5′-GCTCCCCACAGATGGTATCG-3′ RP: 5′-GGAGCCGAGCATATTGACGA-3′	170
9.	*shc2*	NM_001044973.1	FP: 5′-TCGGGCTCAAACTTCACATCT-3′ RP: 5′-GGCCGAATCTACTCCCCTCT-3′	130
10.	*sod2*	NM_199976.1	FP: 5′-TAGGTCTGTTGGTTGGTCGC-3′ RP: 5′-GCACCTAACAGGGGGTTGAA-3′	109
11.	*dnm1l*	NM_200922.1	FP: 5′-GCAGAGTAGCGGGAAGAGTT-3′ RP: 5′-TCCATCCACTCCGTTCTCCT-3′	160
12.	*mfn1*	BC057468.1	FP: 5′-AACTGATGTGACCACCGAGC-3′ RP: 5′-CGTCCCAGCGATTGTTCAAG-3′	179
13.	*mfn2*	NM_001128254.2	FP: 5′-AGACAGTGTTTCGCCCTCAG-3′ RP: 5′-CCTGGTCGTTGTAGCCCATT-3′	163

**Table 3 cells-12-01509-t003:** List of siRNA used in this study.

S. No.	Gene	siRNA Sequence
1.	*il1b*	Sense: 5′-GGAAAGACACCGAGCGCAUUU-3′ Antisense: 5′-PAUGCGCUCGGUGUCUUUCCUU-3′

## Data Availability

Required details have been added in manuscript.

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
