# Peer review of "β-Catenin Elicits Drp1-Mediated Mitochondrial Fission Activating the Pro-Apoptotic Caspase-1/IL-1β Signalosome in Aeromonas hydrophila-Infected Zebrafish Macrophages"

_cells, 2023, doi:10.3390/cells12111509_

Round 1

Reviewer 1 Report

This manuscript could be further considered after major revision. Some questions need to be addressed. As follows:

(1)In the figure 5,autophagy need perform more experiments for endpoint. For examples, authors could perform cell viability or survival rates using Zebrafish.

(2) In the figure 2, authors need to provide flow cytometry analysis for AV/PI.

(3) I also suggested authors provided the results of SiRNA identification using QRT-PCR or protein expression.

Reviewer 2 Report

“Beta-catenin elicits Drp1-mediated mitochondrial fission activating the pro-apoptotic caspase-1/IL-1Beta signalosome in Aeromonas hydrphila-infected zebrafish macrophages” presented by Shibnath Mazumder and coworkers describes the discovery of the canonical Wnt signaling pathway implications in zebrafish macrophages apoptosis after bacterial infection. Authors provide an extensive data set in order to demonstrate how A. hydrophila infection promotes apoptosis by activating canonical Wnt signaling that promotes beta-catenin activation and the subsequently induction of mtROS formation that induces cellular apoptosis. Functionally, authors aim to link the apoptosis induction to the cells capacity to control bacterial infection.

Overall, the investigation is done methodically and detailed, and the conclusions are attractive, however, authors assume specific hypothesis without provide negative controls, that, in mi opinion, could strongly strengthen manuscript quality.

Major points:

1- From my point of view, the major weakness of the manuscript is the assumption of the central role of canonical Wnt signaling pathway in cell death induction by apoptosis after bacterial infection.

First, in Figure 2 authors have to include more controls in order to confirm that cell death after infection is mediated by apoptosis, for example, the addition of a more direct assay (e.g. the western blot analysis of caspase 3, 8 and 9 activation by cleavage) build up the workforce of intrinsic apoptosis pathway induction and discard extrinsic via; and the incorporation of some assays that discard necrosis or necroptosis.

Then, the incorporation of the analysis of the activation of other signaling pathways that could be implicated in apoptosis induction or prevention (e.g. cGAS-STING, JAK-STAT) as negative controls, support the authors hypothesis, since, as is showed in Figure S1, the treatment with Wnt inhibitor JW67 is not enough to block the nuclear beta-catenin concentration. The relevance of this control is outstanding for the conclusion strength. Moreover, the demonstration that other chemicals that interfere with Wnt-beta catenin signaling as LiCl, celecoxib or sulindac, also revert beta-catenin activity, mtROS formation and mitochondrial fission, is important to support JW67-derivated conclusion.

2- Figure legends avoid technical details that are necessary for the result interpretation. Although experimental procedures are detailed in Material and methods section, it is really annoying go back each time to this section after analysis of each figure panel. For example, in Figure 1 is not described how the cytotoxicity is measured and how many cells are analyzed; or in Figure 2 the number of cell analyzed is absent, etc.. Also, in general, pretreatment with several chemicals are used for experiment procedures, but the timing of this pretreatment is not clear in the test.

3- The systematic usage of abbreviation usage is intensely in the whole manuscript, and some abbreviations are not detailed (e.g. NOX, MPTP). The use of “+B” in figure graphs is especially notable, as substitution of MOI and post-infection time. +B helps to vlarify the figure, nevertheless, at least, the first time that authors use the abbreviation, this have to be detailed. In the same way, several chemicals are used in this study, abbreviation and activity are indicated in M&M section, however, the justification of their usage as negative or positive controls is not always indicated.

4- In general the resolution of the figures is limited, specially, figure 2A and 2B, and microscopy panels.

5- Mitochondrial morphology experiment in Figure 6 is not quantitative analyzed and only one, of the three recorded fields, is showed. Authors do not indicate the total number of cells captured in each field. The quantification cell by cell of the experiment is necessary, showing the total number of cells that display the phenotype respect to the total cells analyzed, since one of the major conclusion of the work is the implication of mitochondrial fission in apoptosis induction by bacterial infection. Specifically, authors could quantify cell by cell, using an appropriated software, as the free software Fiji/ImageJ, the mitochondrial morphology by measuring the total of positive mitotracker particles detected in each cell and measuring their areas and intensity.

6- Antimycin A treatment is used as positive control for mtROS production in Figure 5, surprisingly, however, this positive control is avoided in the following figures. The incorporation of this chemical helps to clarify that the phenotypes observed depends on mtROS production. The requisite to use Ant A is especially relevant for the figures that supports major conclusions as caspase activation, mitochondrial fission induction by Drp1, and the demonstration of pro-apoptotic and anti-bacterial role of beta-catenin.

Minor comments:

1- Figure S3 shows a time course assay for sod2 fold change after bacterial infection, while Figure 5C exclusively shows the relative fold change at 12h p.i., it could be interesting to repeat this experiment in a time course way, since sod2 repression kinetics could be informative.

2- Figure 6A shows a plot line representation instead of columns. Figure 6A is not a time course experiment, and the plot line that connects each analyzed sample has not sense.  Moreover, YCG063 and CsA are used as positive and negative controls, conversely, the relative change in fluorescence is not statistically significant. Finally, vertical bars are not centered respect the nodes and subsequently figure interpretation is weird. The change for a column representation is strongly suggested, and the differences between positive and negative control have to be revised.

3- Figure 8 and Figure S5 could be used as a support for apoptosis induction in the first part of the manuscript.

4- The inclusion of a table that summarize the chemicals used for cells treatment, their molecular targets and the employ as negative or positive control in each figure, is also toughly suggested.

In summary, the manuscript presented by Mazumder and coworkers includes several experimental approaches that support author conclusion, nevertheless, the final quality of the article could be improved in three ways:

1) The insertion of negative controls that support the conclusion that cell death induction after bacterial infection is promoted specifically by cell death stimulation by canonical Wnt pathway activation apoptosis induction.

2) The description in more precise detail figure legends in order to support experiment analysis, and the revision of the quality and resolution of the figures.

3) The microscopy result of mitochondrial fission should be reexamined by using a quantitative and conscientious approach.

Round 2

Reviewer 1 Report

Authors have addressed our questions. Current version could be accepted .

Reviewer 2 Report

The revised manuscript version presented by the author mostly answered all my concerns. The improvement of microscopy analysis, the clarification of controls and the inclusion of new controls strongly reinforce the manuscript quality. However, I’m not agree with author respect to time course experiments, independently of the selection of the more precise time point to show the strongest differences, the kinetic of the cellular processes studied in this paper could be extremely informative. Nevertheless, the data presented support the conclusions of the work. So, in my honest opinion the article should be accepted.